# The Ubiquitin–Proteasome System (UPS) and Viral Infection in Plants

**DOI:** 10.3390/plants11192476

**Published:** 2022-09-22

**Authors:** Dania P. Lobaina, Roberto Tarazi, Tamara Castorino, Maite F. S. Vaslin

**Affiliations:** 1Laboratório de Virologia Molecular Vegetal, Departamento de Virologia IMPPG, Universidade Federal do Rio de Janeiro, Rio de Janeiro 21941-599, RJ, Brazil; 2Programa de Pósgraduação em Biotecnologia Vegetal PBV, UFRJ, Rio de Janeiro 21941-971, RJ, Brazil

**Keywords:** ubiquitin proteasome system, plant viruses, 26S proteasome

## Abstract

The ubiquitin–proteasome system (UPS) is crucial in maintaining cellular physiological balance. The UPS performs quality control and degrades proteins that have already fulfilled their regulatory purpose. The UPS is essential for cellular and organic homeostasis, and its functions regulate DNA repair, gene transcription, protein activation, and receptor trafficking. Besides that, the UPS protects cellular immunity and acts on the host’s defense system. In order to produce successful infections, viruses frequently need to manipulate the UPS to maintain the proper level of viral proteins and hijack defense mechanisms. This review highlights and updates the mechanisms and strategies used by plant viruses to subvert the defenses of their hosts. Proteins involved in these mechanisms are important clues for biotechnological approaches in viral resistance.

## 1. Introduction

In the last thirty years, the ubiquitin–proteasome system (UPS) has become a relevant issue in biology. Currently, the UPS plays a central regulatory process in virtually all aspects of eukaryotic cells [1]. The cell’s very existence is only possible with the proper functioning of the UPS and its participation in all vital processes, from the maintenance of cellular genetic integrity, protein production, signaling, transport, differentiation, and survival. In plants, the UPS acts in germination, flowering, senescence, and the response to abiotic and biotic stress, reviewed by [2,3,4]. The UPS’s primary function is protein degradation in the 26S proteasome. In addition, the UPS components have an essential role in plant–pathogen interactions, acting in fundamental defense mechanisms, as extensively shown. This review exposes some of the most remarkable facts within the plant viral universe versus the host’s UPS, highlighting important key proteins that may be putative biotech candidates for virus resistance improvement.

## 2. The 26S Proteasome

The 26S proteasome is responsible for degrading ubiquitinated proteins [5], reviewed by [6]. The 26S proteasome is a 2.5 MDa multisubunit protease located in the cytosol and nucleus of every cell throughout the eukaryotic kingdom. It contains two sub-complexes with well-differentiated functions. One sub-complex is the 20S core particle (CP), in the form of a barrel where protein degradation occurs. The other, called the 19S regulatory particle (RP), is responsible for capturing and preparing the substrate to enter the degradation site correctly. The CP has a hollow cylinder shape of four rings, one on top of the other. α-rings have seven subunits (α1−7) and are located at the cylinder ends, while the two rings with the β subunits (β1−7) occupy the cylinder’s central part [7,8]. Lid and base components compose regulatory particle 19S. The lid, which has nine subunits, recognizes ubiquitinated substrates and removes ubiquitin (Ub) chains. The base has several subunits and unfolds the substrate. RP recognizes ubiquitinated substrates, unfolding them and translocating them into the CP [9,10,11]. Advances in X-ray, crystallographic imaging, and cryo-electron microscopy (EM) enabled the creation of an exceptionally well-resolved proteasome model and are responsible for its current understanding [10,11,12,13,14,15,16] (Figure 1).

## 3. Ubiquitinylating Enzyme Cascade

Over 6% of the predicted *Arabidopsis thaliana* genome encodes UPS proteins, including only two E1, 37 predicted E2 proteins [17], and at least 1400 predicted E3 proteins [17,18]. Four subfamilies compose the E3 Ub ligases: HECT (homologous to the E6-AP carboxyl terminus), RING (really interesting new gene), U-box, and CRL (cullin-RING ligase). The HECT, RING, and U-box E3 ligases are single polypeptides, whereas the CRLs consist of multiple subunits [19,20,21]. At this moment, CRLs are likely the best-characterized E3s in plants [22]. CRLs participate in almost all aspects of plant growth and development, auxin signaling, JA signaling, and regulation of antiviral response [23,24,25,26,27,28,29,30]. In the CRL complex, the cullin protein serves as an elongated scaffold, recruiting the RING box protein 1 (RBX1) and E2 to its carboxy (C)-terminal region and binding a substrate adaptor to its amino (N)-terminal region. There are four subtypes of CRLs in plants, each with a different cullin: CUL1, CUL3, CUL4, and the cullin-like protein anaphase-promoting complex 2 (APC2). CUL1 E3 ligases, also called SCF from S-phase kinase-associated protein 1 (SKP1)–CUL1–F-box complexes, are the most carefully studied class in which the substrate adaptor consists of SKP1 and an F-box protein [31,32]. SCFs are essential for cell-cycle regulation and hormone signaling in plants and are closely related to many cellular processes in eukaryotes (reviewed by [33]). CUL3 E3 ligases regulate cellular processes such as biosynthesis, hormone signaling, and response to light and stress (reviewed by [34]).

The most common families of E3 Ub ligases impacting plant–pathogen interactions are the RING and U-box E3 Ub ligases [35,36]. RING and U-Box predominate in the host-pathogen relationship, and both transfer Ub from the E2-Ub complex to the substrate. Most functionally characterized RING and U-boxes are found in the cytosol and throughout the endoplasmic reticulum (ER) and participate in ER-associated degradation (ERAD). A recent report of a genome-wide analysis of U-box ligases of tomatoes confirms that E3 ligases, which ultimately mark proteins, provide substrate specificity in the UPS [36]. Each E3 ligase family controls the binding of ubiquitins to only one or a small subset of substrate proteins, which accounts for the large number of genes that comprise the E3 ligase family. There are 508 RING domains according to the annotated *Arabidopsis* genome [37]. The growing interest in these topics reflects the constant flux of research and publications in crops of interest like wheat (*Triticum aestivum*) [38] and flax (*Linum usitatissimum*) [39].

## 4. The Ubiquitin (Ub)

Like other UPS components, ubiquitin (Ub) has undergone very little diversification and remained almost constant throughout evolution in organisms in the eukaryotic kingdom [40,41]. It is currently one of the most important post-translational modifications to maintain cellular homeostasis [42]. Three enzymes work in a cascade to attach ubiquitin covalently to the substrate. It begins with the activation of Ub by E1 or the ubiquitin-activating enzyme; it continues with E2 or ubiquitin-conjugating enzyme, and ends with E3 or ubiquitin ligase [7,8] (Figure 1).

In the initial ATP-consuming reaction, E1 first activates the ubiquitin moiety by forming a high-energy thioester bond between an E1 Cys residue and the carboxy-terminal Gly of ubiquitin. This activated ubiquitin is then donated to a Cys on an E2 by trans-esterification. In most cases, the ubiquitin-E2 intermediate serves as the proximal ubiquitin donor, using an E3 to identify the target and catalyze ubiquitin transfer. The final product is a ubiquitin conjugate. The C-terminal Gly carboxyl group of ubiquitin is linked through an isopeptide bond to an accessible amino group (typically a Lys ε-amino) in the target [4]. By attaching ubiquitin in various ways, it is possible to create distinct target fates. This binding of Ub molecules to the substrate is reversible through deubiquitinase enzymes (DUBs), described for the first time by [43].

A substrate protein of the ubiquitin system may be (i) monoubiquitylated, whereby a single ubiquitin molecule conjugates to the substrate; (ii) multimonoubiquitylated, resulting in the conjugation of several ubiquitin molecules to the same substrate protein on different lysine residues; or (iii) polyubiquitylated, in which additional ubiquitin molecules are conjugated to the first ubiquitin molecule, resulting in the formation of a polyubiquitin chain on one lysine residue of the substrate protein. The shortest polyubiquitin chain capable of activating degradation by the 26S proteasome is four monomers [7].

## 5. Ubiquitin-Proteasome System (UPS) Role under Viral Infection

The UPS acts ambiguously during viral infections. In some cases, the UPS can improve the function of viral proteins by ubiquitin adding as a post-translational modification. On the other hand, the UPS can degrade viral proteins as a host defense mechanism to counteract infection [44]. Remarkably, plant viruses can manipulate UPS for their benefit. The various means range from inducing, inhibiting, or modifying the enzymes involved in the process, especially the E3 ligases. Some viruses such as turnip yellow mosaic virus (TYMV) and maize rayado fino virus (MRFV) can encode deubiquitinase enzymes (DUBs) to achieve a favorable environment for viral infection by inhibiting host defense mechanisms [45,46].

Two viruses belonging to the *Tombusvirus* genus, tomato bush stunt virus (TBSV) and cymbidium ringspot virus (CymRSV), help elucidate the plant’s UPS during viral replication [47]. Replication in the infected cell occurs after translating two replication proteins: p33, an RNA chaperone, and p92^pol^, an RNA-dependent RNA polymerase (RdRp). Two subgenomic RNAs also originate from genomic RNA to express three viral proteins related to cell-to-cell movement, viral particle assembly, and gene silencing suppression. Several studies using the yeast protoarray system and genome-wide screens have shown many host genes linked to viral replication in this group of viruses [48,49]. Ubiquitin has a fundamental role in *Tombusvirus* infection. In infected plants, the copurification of TBSV p92^pol^ replicase with the Arabidopsis UBC2 (ubiquiting-conjugating enzyme 2) and CDC34p (cell division cycle protein 34) E2 ubiquitin-conjugating enzymes are essential for replicating tombusviruses [47]. UBC2 and CDC34p participate in cellular processes and can ubiquitinate p33 with one or two molecules. Ubiquitination enables the recruitment of ESCRT (endosomal sorting complexes required for transport) proteins bound to Vps23p involved in membrane flexion and invagination and the formation of viral spherules during tombusvirus replicase complexes [50,51].

Tombusvirus replication ubiquitinates lysines K70 and K76 of p33. Both are monoubiquitinated or biubiquitinated, and together with a sequence similar to a late domain (sequence P T/S XP, where X symbolizes any amino acid), Ub-lysines facilitate the binding between p33 and the host factor Vps23p ESCRT-I. Mutations in both lysines and/or in a sequence similar to the late domain of p33 disturbed the interaction with Vps23p, causing a decrease in TBSV replication in yeast and plant cells [51].

Studies related to the role of host proteins in the interaction with TBSV viral proteins demonstrated the function of Rsp5p. Rsp5p is an E3 ligase member of the Nedd4 family, with three types of functional domains. Rsp5p was shown to bind to p33 and p92 and is a negative regulator of TBSV replication. The overexpression of Rsp5p, which has a WW protein interaction domain and a protein-ubiquitination domain HECT, decreases the level of viral replication of TBSV and vice versa [52]. Additionally, the WW domain plays a crucial role in inhibiting replication and the endosome–vacuole pathway in a proteasome-unrelated mechanism [52]. Another UPS host protein is Rpn11p (regulatory particle non-ATPase), a deubiquitinase that forms a heterodimer with Rpn8p. Both are part of the lid of the regulatory particle (RP), one of the components of the 26S proteasome [14]. Rpn11p is known to intervene in the TBSV viral replication complex (VRC), where it interacts with the viral replication protein p92 and recruits the DDX3-like Ded1p/RH20 DEAD-box helicase, a potent suppressor of the viral recombination. Rpn11p has a dual function because it intervenes in the proper assembly of the proteasome. It is also a key player in the assembly of the VRC of TBSV, influencing the virus’s replication and genetic recombination [15,53].

The presence of the P3 protein of the *Rice grassy stunt virus* (RGSV), a negative-sense single-stranded RNA virus, induces an increase in ubiquitination and degradation of the rice nuclear protein RNA polymerase D1a (OsNRPD1a) [54]. OsNRPD1a is one of the two orthologs of RNA polymerase IV (Pol IV) required for RNA-directed DNA methylation (RdDM). The accumulation of P3 induces the expression of host P3-inducible protein 1 (P3IP1), a U-box E3 ligase that interacts with OsNRPD1a, marking it for degradation. Interestingly, the *OsNRPD1* knockdown and the overexpression of *P3IP1* in rice produce symptoms similar to RGSV infection. The absence of OsNRPD1-driven methylation generates smaller plants with development deficiencies. These findings point to a new mechanism where RGSV induces degradation of host RNA Pol IV polymerase, facilitating infection and producing the development of disease symptoms. Therefore, P3IP1 degradation of OsNRPD1 via the proteasome increases RGSV susceptibility.

## 6. Viral Movement Proteins and UPS

The UPS and movement protein (MP) interaction was studied using the turnip yellow mosaic virus (TYMV, genus *Tymovirus*). The translation of the genomic RNA of TYMV produces MP 69K and polyprotein 206K, composed of proteins 66K and 140K. The 66K protein, a component of RNA-dependent RNA polymerase (RdRp), is degraded via the proteasome, causing its decrease and affecting viral replication. Besides RdRp, the MP 69K is also degraded by 26S proteasome, with the polyubiquitination of lysines 109 and 111 [55]. The degradation of MP can have two interpretations. In one way, the UPS may protect cell homeostasis, reducing foreign potentially toxic proteins that can be harmful to the cell. This idea makes sense, since MPs facilitate the passage from one cell to another through the plasmodesmata, so the degradation of these viral proteins would be a way to protect the cellular integrity of the host and thus guarantee the survival of the cells. Interpreted in another way, this degradation of viral MPs constitutes a defense mechanism of the host to reduce the spread of the virus to neighboring cells [55,56].

The best well-studied MP is the 30K MP from the tobacco mosaic virus (TMV, genus *Tobamovirus*). Almost 22 years ago, Reichel & Beachy (2000) [56] showed that the 26S proteasome degrades MP 30K from the mobile complexes associated with the endoplasmic reticulum (ER). In the presence of MP, mobile complexes associated with ER constitute virus “factories” where TMV replication takes place. However, in the late stages of infection, the accumulation of MP in the cell hinders normal functioning, damaging the cortical ER and the cytoskeleton. The cell only recovers its functions when the MP has been degraded [56]. In the presence of a proteasome inhibitor, an increase and intercellular accumulation of MP 17K of the potato leaf roll virus (PLRV, genus *Polerovirus*) occurs [9]. These phenomena observed in the TYMV, TMV, and PLRV demonstrate that viruses can modify UPS functions through various strategies to achieve cell survival on their own. More in-depth studies are needed to characterize each component of UPS involved in MP degradation and the role of each component in the face of attack by phytopathogenic viruses.

## 7. Unfolded Protein Response and UPS

The unfolded protein response (UPR) is a protein quality-control system in the endoplasmic reticulum that routes defective proteins for degradation in the proteasome and prevents their accumulation to dangerous levels for cellular balance. There is evidence that in the absence of bZIP60, a transcriptional factor involved in the UPR, potato virus X (PVX) cannot successfully infect protoplasts or plants [57]. PVX TGBp3 movement protein resides in ER and activates the upregulation of SKP1. These facts indicate that the UPR and UPS are necessary for regulating damaged proteins and maintaining tolerable PVX TGBp3 levels. In the absence of UPR, PVX TGBp3 can accumulate toxic levels and prevent successful virus infection [57].

## 8. The Role of the UPS in Plant Immunity

The UPS mediates the degradation of cellular proteins when viruses interact with the host cell. Inhibition of SGT1, a highly conserved component of the SCF E3 and COP9 signalosome (CSN), a multiprotein complex involved in degradation, abolishes the resistance mediated by the N gene against TMV, thus suggesting a pivotal role for the UPS in regulating the innate immune response of plants [58].

As part of the SCF complex, SNIPER7 (snc1-influencing plant E3 ligase reverse) regulates the response to pathogen attacks on plants. SNIPER7 interacts with the CDC48 (cell division cycle 48) unfoldase as part of the SCF E3 ligase complex, marking it to CDC48 for degradation via the proteasome pathway. Given the large number of functions that CDC48 fulfills in the cell and considering that its absence is lethal, the interaction with SNIPER7 is finely regulated [59]. SNIPER4 (snc1-influencing plant E3 ligase reverse) regulates two redundant tumor necrosis factor receptor (TNFR)-associated factors (TRAF) proteins, called MUSE13 (mutant, snc1-enhancing 13) and MUSE14, related to the immune response in plants. MUSE13 and MUSE14 function as adapters between the F-box CPR1 and its substrates nucleotide leucine rich immune sensors suppressor of NPR1, constitutive 1 (SNC1) and resistant to *Pseudomonas syringae* 2 (RPS2). The lack of MUSE13 and MUSE14 causes extreme autoimmune behavior due to the accumulation of SNC1, which interferes with plants’ proper growth and development [60].

E3 ligase SAUL1 (senescence-associated E3 ubiquitin ligase 1) is a positive regulator of PAMP-triggered immunity, and the immune receptor SOC3 (genetic suppressor of CHS1-2-3) controls homeostasis [61]. Other E3 ligases, members of the HECT family, play a fundamental role in the immune response generated by plants. The salicylic acid (SA) signaling pathway is significantly affected when mutating UPL1 (ubiquitin–protein ligase), UPL3, and UPL5 proteins. Mutants of upl3 proved unable to restart SA-induced transcriptome formation and failed to orchestrate defense against hemibiotrophic pathogens. UPL3 is essential to generate an immune response in plants to the attack of pathogens and interacts with the proteasome’s regulatory particle and other UPS components [62].

## 9. Viral Protein Degradation by the UPS

The degradation process in the 26S proteasome depends on the activation of an enzymatic cascade composed of E1-E3 enzymes. Among the critical factors for pathogenesis is the ability of viruses to subvert host pathways. One of the main sites where this occurs is UPS. Viruses have coevolved to use the host’s UPS to their advantage in many aspects of their life cycle, including the exit of the virus from the invaded cell, the increase in viral replication, the alteration of the cell cycle, and evasion from defense mechanisms of the host [63]. The regulation of the UPS is critical for the correct operation of the defense systems of plants. Therefore, many pathogens, including viruses, have evolved to evade or counteract these mechanisms.

In the interaction between the virus and host E3 ligases (and their regulators), viruses hijack SCFs and CRL, influencing E3 ligase activity, degrading host proteins, and facilitating viral spread [31,32,64]. Furthermore, some viruses have a very particular way of evading the defense response of plants by encoding F-box proteins. This uniqueness allows them to modify the functioning of E3 ligases to benefit the viral machinery, increasing viral replication capacity (Figure 2). Tomato chlorosis virus (ToCV) p22 protein have a motif F-box like, can suppress the auxin signaling pathway by competing with NbTIR for binding to the C-terminal domain of NbSKP1, which interferes with the assembly of the SCFTIR E3 ligase complex, increasing the accumulation of viral RNA and the severity of symptoms [65] (Figure 2b). There is evidence of different strategies among criniviruses to subvert the UPS’s functioning through interference in the SCF E3 ligase complex [66]. Overexpression of the C4 of beet severe curly top virus (BSCTV, genus *Geminivirus*) protein is decisive in the virus’s leading symptoms and generates an atypical host cell division. RKP, a RING finger protein, is induced by BSCTV C4. RKP is a functional E3 ligase and interacts in vitro with cell cycle inhibitory ICK/KRP proteins, which accumulate when RKP is mutated. In the presence of BSCTV, the level of ICK/KRP decreased, but with overexpression, the susceptibility to BSCTV infection decreased (Figure 2b). The induction of RKP by the BSCTV C4 protein can affect virus infection by regulating the host cell cycle through its interaction with the inhibitory proteins ICK/KRP [67].

The UPS regulates signaling pathways and hormones during viral invasion [68]. The βC1 protein, a multifunctional pathogenicity factor encoded by the satellite DNA of the cotton leaf curl Multan virus (CLCuMuV), subverts ubiquitination to enhance virus infection [69,70]. The interaction of βC1 with S1UBC3 (ubiquitin-conjugating enzyme) correlates with the severity of symptoms in plants, which is similar to the phenotype observed when disturbing the UPS [71]. Interestingly, βC1 interacts with glyceraldehyde-3-phosphate dehydrogenase GLCA1 and 2, inducing autophagy [72]. The above-described behavior reveals how the UPS may regulate signaling pathways and hormone.

Recent publications help to delve into the interaction between polerovirus silencing suppressor protein (P0) and argonaute 1 (AGO1) and its relationship with the SCF complex. One of the most exciting findings is that the brassica yellow virus (BrYV) P0^Br^ can avoid degradation by mimicking the F-box domain and interacting with SKP1, a component of the host’s SCF E3 ligase complex, forming the SCF–P0 complex [73] (Figure 2b). AGO is the most critical enzyme of the RISC complex and is essential for RNA silencing virus defense. Stabilization of P0 in the SCF-P0 complex triggers AGO degradation, eliminating the silencing of the viral RNA, which can lead to successful infection [73]. The P0 F-box domain of an Argentinian potato leafroll virus interacts with *Solanum tuberosum* SKP1 orthologue (StSKP1), triggering the ubiquitination and subsequent degradation of AGO1 and other ARGONAUTE proteins [74].

P0s from the polerovirus beet western yellows (BWYV), cucurbit yellow aphid virus (CABYV), and others and from the Enamovirus pea enation mosaic *virus (PEMV)* are known potent suppressors of RNA silencing, one of the plants’ primary defense strategies [75,76]. BWYV and CABYV P0s interact with *Arabidopsis* SKP1 homologs (AtSK1 and AtSK2). The suppressive activity of P0 silencing was already related to the interaction of its F-box domain with the AGO1 PAZ motif and adjacent sequences of AGO proteins to mediate its degradation [75,77]. However, P0-mediated AGO1 degradation occurs by autophagy, an independent proteasome pathway [76]. Li et al. [73] recent findings show that the stabilization of P0 by forming the SCF-P0 complex is crucial for preventing its degradation by proteasome and enabling virus infection.

The first report of a virus-encoded F-box interacting with a host component was founding in bean necrotic yellow virus (FBNYV, genus *Nanovirus*). One of the proteins encoded by component 10 of FBNYV, C10, called clink, for “cell cycle link”, contains an F-box that binds to SKP1 from *Mendicago sativa*. Clink was also found to interact with the cell cycle regulator retinoblastome (RB) protein, increasing viral replication capacity [78] (Figure 2b). Upon RB activity being altered, the virus acquires the ability to modify the cell cycle and force DNA synthesis, creating a favorable cellular environment for the successful replication of the viral genome.

## 10. Viral Strategy Using E3 to Protect Unstable Proteins

An inverse viral strategy is the modification of the UPS to stabilize cellular proteins that are very unstable under normal conditions. The interaction of the geminivirus BSCTV C2 protein with the *Arabidopsis* S-adenosylmethionine decarboxylase proenzyme 1 (SAMDC1) enzyme inhibits the degradation of SAMDC1 through the 26S proteasome. Thus, host methylation pathway suppression and host reduction de novo methylation in the presence of BSCTV C2 favors the replication of viral DNA [79] (Figure 2b). Moreover, C2 from other geminiviruses, such as tomato yellow leaf curl virus (TYLCV) and beet curl virus (BCTV), can modulate the function of some ligases of the SCF complex of infected cells, inhibiting host defense JA pathway activation [80]. Transgenic *Arabidopsis* plants expressing C2 suppressed the response to the hormone jasmonate (Figure 2b). Jasmonic acid and its metabolites, called jasmonates (JA), integrate relevant signaling pathways in plants and are part of various physiological processes and responses to biotic and abiotic stress. Under basal conditions, JA levels are low, and jasmonate ZIM-domain (JAZ) proteins repress the expression of related genes. The expression of bioactive jasmonate (JA-Ile) increases with stressful elements, which mediates the interaction between the JAZ repressors and the F-box protein coronatine-insensitive 1 (COI1), a member of the receptor complex from JA, the E3 ligase SCFCOI1. These relationships allow JAZ to be ubiquitinated and subsequently degraded via the 26S proteasome, allowing the expression of JA-responsive genes. This behavior makes sense if we think that the jasmonate signaling pathway begins when components of the SCFCOI1 complex detect the hormone [81].

The expression of C2 directly affects the plant’s response to viral infections because it influences the activity of the CSN, one of the regulators of the activity of CRL E3 ligases. CSN is in charge of removing the remains of the related to ubiquitin (RUB) protein (also called NEDD8). This ubiquitin-like protein binds reversibly to cullin 1 (CUL1) from CRL ligases [82,83]. CUL1 is part of the SCF complex and its neddylation upregulates CRL activity. If this release of RUB does not occur, CUL 1 accumulates in the cell in its rubylated form. This accumulation alters the functioning of the cell because of the incorrect regulations made by CRL ligases involved in the activation of various hormones involved in defense responses. Hence, C2/L2 is a critical powerful viral strategy to achieve successful infection [80].

SCF complexes regulate many physiological processes, some with a recognized pleiotropic character that complicates the interpretation of phenotypes [84]. Therefore, the capability of geminiviruses to encode proteins that intervene in SCF functions is a relevant tool that can intervene in most of these processes that maintain the proper development of life in plants; specifically, those related to the hormonal defense response, as is the case of jasmonate, the main target of tomato yellow leaf curl Sardinia virus (TYLCSV) C2 protein through interference with CSN and, by extension, of the SCF complex [80] (Figure 2b).

No less surprisingly, some SCF complexes escape this effect by overexpressing the F-box, which also occurs during geminivirus infection [85]. These findings suggest that geminiviruses selectively intervene in the modulation of the subunits of the SCF complex, both negatively and positively.

Furthermore, the C1 protein of the DNA β of the CLCuMuV also interacts with the ubiquitin-conjugating enzyme of tomato SlUBC3, and the presence of the myristoylation-like motif domain of βC1 is necessary for the development of viral symptoms [69]. βC1 CLCuMuV regulates the host ubiquitination pathway through its interaction with SKP1 in *N. benthamiana* (NbSKP1) [70]. This interaction leads to the disruption of the binding of NbSKP1 with NbCUL1. When silencing *NbSKP1* or *NbCUL1*, viral genomic DNA accumulates, and symptoms increase substantially. These results confirm that βC1 from CLCuMuV inhibits the process that the SCF E3 ligase complex should carry out through interaction with NbSKP1, which promotes an increase in viral infection and the induction of more severe symptoms in the host [70] (Figure 2b).

Autophagy is an immune response of plants that geminiviruses strategies are capable of subverting. The selective autophagic receptor NbNBR1 improves expression in the presence of βC1 from tomato yellow leaf curl China betasatellite (TYLCCNB) associated with the tomato yellow leaf curl China virus (TYLCCNV). The interaction of NbNBR1 and βC1 produces cytoplasmic granules and protects βC1, a substrate for the RING-finger NbRFP1 E3 ligase, from degradation. The overexpression of NbNBR1 in *N. benthamiana* increases the accumulation of βC1 and favors viral infection (Figure 2b). In contrast, viral infection inhibition happens when NbNBR1 is mutated or silenced, lowering βC1 levels [86]. These results demonstrate the ability of TYLCCNB and TYLCCNV to protect themselves from the action of the E3 ligase NbRFP1 through the creation of cytoplasmic granules formed from the interaction between the viral protein βC1 and NbNBR1, an autophagic protein.

## 11. Conclusions

The UPS performs a regulatory function and is closely related to essential processes carried out by the cell, including the activation of defense mechanisms. However, viruses developed different strategies to successfully suppress or modulate the UPS to establish infection. Currently, numerous advances in the field of technology allow a more in-depth study of the complex interactions that take place between the UPS of the host and phytopathogenic viruses. Viral strategies to escape the host’s defense mechanisms deserve special mention, ranging from the hijacking of central components of the UPS such as the presence of genes that encode proteins that can compete with host proteins and “trick” the UPS, managing to subvert hormonal signaling mechanisms, key in plant defense. The weak interactions between E3 ligases and their known substrates demonstrate the need to create new tools, both in vitro and in vivo, that contribute to the discovery and characterization of still unknown substrates, as well as the extensive and complicated regulatory network that governs the behavior of the UPS against the attack of pathogens. Finally, designing new plant cultivars resistant to the most devastating viral diseases will need in-depth studies of the interaction between host viruses and the UPS.

## Figures and Tables

**Figure 1 plants-11-02476-f001:**
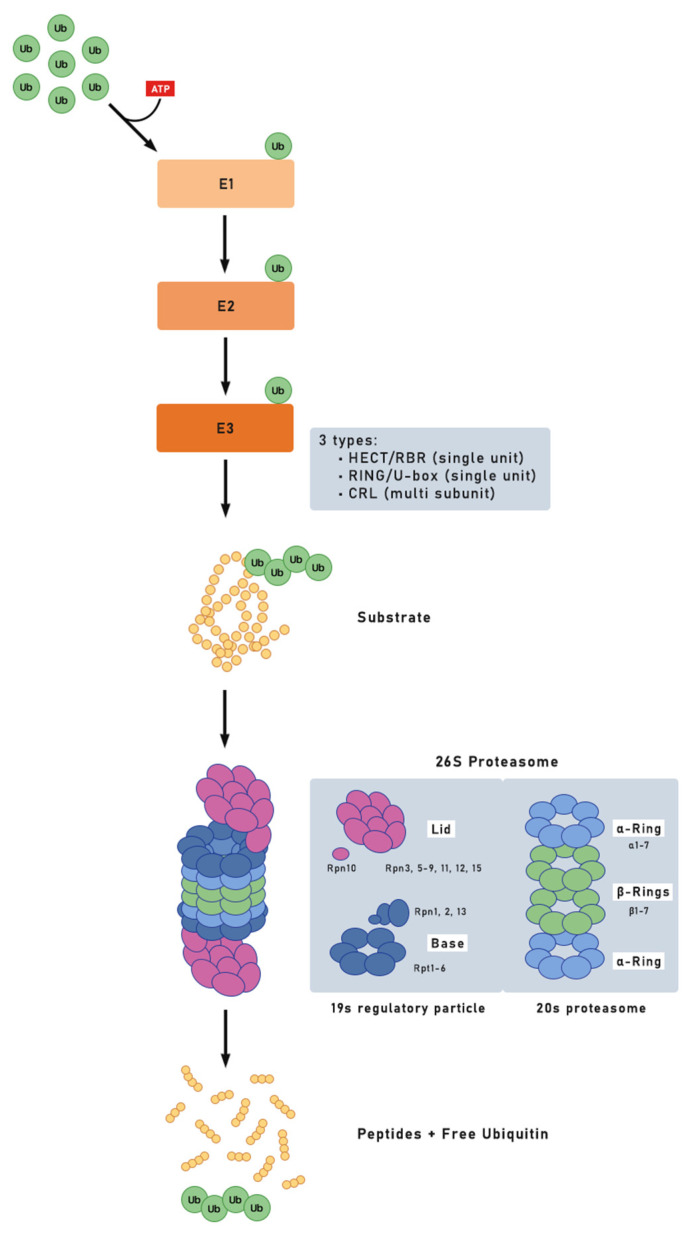
E1 activates a ubiquitin molecule (Ub) in an ATP-dependent reaction by forming a high-energy thioester bond between an E1 Cys residue and the carboxy-terminal Gly of ubiquitin. This activated ubiquitin is then donated to a Cys on an E2 by trans-esterification. In most cases, the ubiquitin–E2 intermediate serves as the proximal ubiquitin donor, using an E3 to identify the target and catalyze ubiquitin transfer. The final product is a ubiquitin conjugate. The C-terminal Gly carboxyl group of ubiquitin is linked through an isopeptide bond to an accessible amino group (typically a Lys ε-amino) in the target. Depending on what lysine was ubiquitinated, mostly K11 and K48, proteasome 26S could degrade the target protein.

**Figure 2 plants-11-02476-f002:**
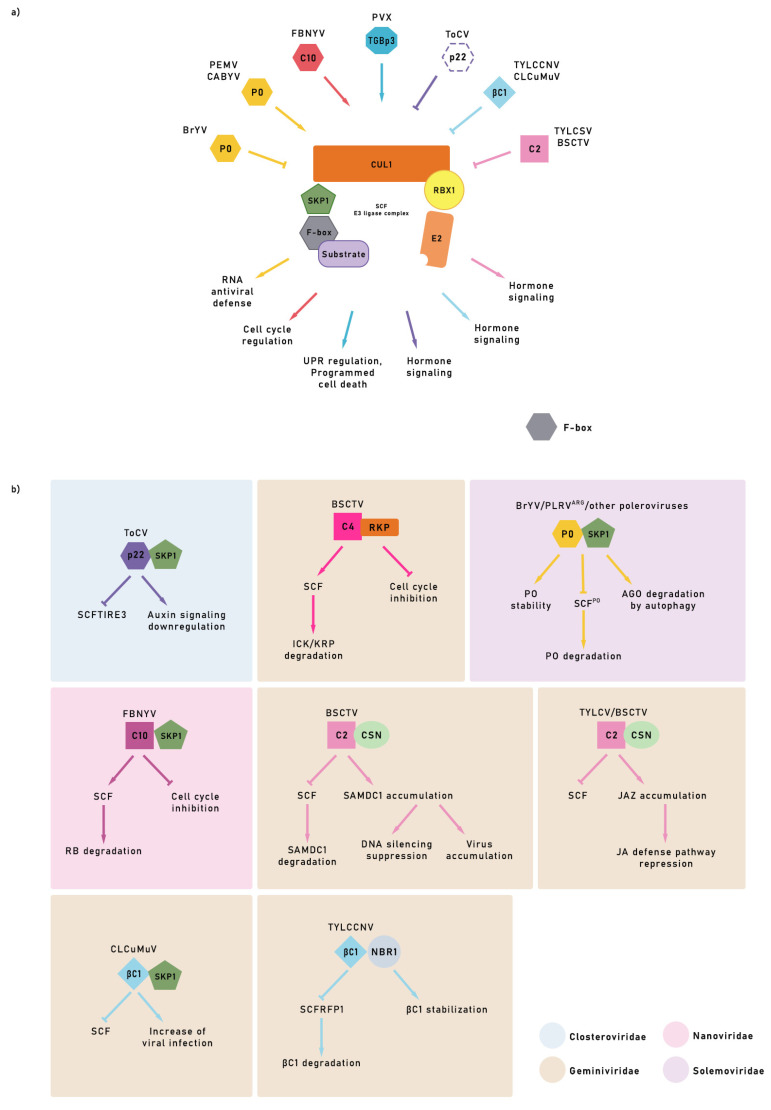
(**a**) Examples of viral proteins subverting SCF E3 ligase complex. Several viruses had already been shown to encode proteins that may interfere or plays as components of the SCF E3 ligase complex to benefit viral infection. Color of the arrows are representing the cellular process affected by the subverting of E3 ligase complex by individual virus. Viral proteins represented by a hexagon shape may act as plant F-box proteins taking part directly of the complex, leading to the ubiquitination of their targets, or may compete for SKP1 with the host F-box protein. A putative host F-box protein is shown in gray inside the complex. (**b**) Some of these viral proteins may hypothetically take part in the complex acting as F-box proteins, as already shown for P0, C10, and p22. P0 of the polerovirus interacts with host SKP1, affecting RNAi antiviral defense. BrYV P0 interaction with SKP1 in *A. thaliana* suppress P0 degradation by SCFP0 stabilizing it and allowing P0 AGO degradation by autophagy. Clink (C10) of FBNYV interacts with SKP1 and with RB. As C10 interferes in cell cycle regulation, it can be hypothesized that it may target RB to be ubiquitinated by SCF complex allowing cell cycle activation. The p22 protein of ToCV presents an F-box-like domain that interacts with host SKP1 and interferes with the correct assembling of host SCF complex, perturbing hormone signaling. The C4 protein of the geminivirus BSCTV interacts with RKP E3 ligase, promoting the degradation of the cell cycle inhibition protein ICK/KRP and inducing cell cycle that favors virus replication. C2 proteins of BSCTV, TYCLV, and BCTV interact with the CSN enzyme, which promotes deneddylation of CUL1 and makes the SCF complexes responsible for SAMDC1 and JAZ degradation unable to work properly. In consequence, DNA methylation and de novo methylation and JA defense pathway are impaired, respectively. The presence of TGBp3 from PVX, by its side, induces host SKP1 expression, affecting UPR regulation and programmed cell death (**a**). The βC1 protein of satellite virus of the geminivirus CLCuMuV interacts with SKP1 interfering with SCF formation. βC1 from TYLCCNV interacts with the NBR1 protein, impairing the correct functioning of SCF RFP1, which would otherwise ubiquitinate βC1 and promote its degradation by proteasome. Thus, interference or usurpation of SCF E3 ligase complex seems to be common among plant viruses.

## Data Availability

Not applicable.

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
