# Peer review of "The Ubiquitin–Proteasome System (UPS) and Viral Infection in Plants"

_plants, 2022, doi:10.3390/plants11192476_

Round 1

Reviewer 1 Report

OVERVIEW: In this manuscript, the authors summarize the role of UPS in viral infection of plants. It would be more logical to change the order of the introductory chapters (2., 3., 4. -› 3., 4., 2) and introduce the UPS from ubiquitin through the enzymes to proteasome. I found disturbing and I see no reason for writing the full name of some proteins or enzymes all with capital letters. Please, change them to small letters capitalizing each word only. Abbreviations can stay all in capital letters.

SPECIFIC COMMENTS:

Line 43-44: '20S core protease' should be changed to '20S core particle'

Line 62: 'before proteasome protein degradation' change to 'to attach ubiquitin covalently to the substrate'

Figure 1: Some modifications would improve the figure. Please, change 'Target' to 'Substrate'. There should be not only one but four to six green ubiquitin unit on the yellow substrate protein. It would be better to position this ubiquitin chain on one of the loops rather than at the terminus of the substrate. On the other side of the proteasome, the degraded peptides should be resembled by three to four yellow dots not only single ones. The cut off ubiquitin chain should be shown still as a set of four to six green ubiquitin units.

Line 94'Ubiquitin-Proteasome System (UPS) proteins' change to 'Ubiquitinylating enzyme cascade'

Overall, the authors present interesting data and correlations of viral proteins and UPS elements that have the potential to contribute to the field. I support the publication of the manuscript after completing the above suggested corrections.

Reviewer 2 Report

The present review describes the role played by UPS under viral infection. The paper seems complete and well detailed, altought some paragraphs appear a bit messy, stuffed with many information.

The design of some additional schematic representations are needed to better summarize the main concepts dealt with, thus favoring a better understanding  of the paper:

Regarding line 53 and 54, please add a graphical rapresentation or x-ray structure of the proteasome model, highlighting the aforementioned regions and proteins structures, to guide the comprehension of this paragraph.

Also for chapter 9 and 10, please add a graphic representation/cartoon of the mechanisms and pathways described. 

Round 2

Reviewer 2 Report

The Authors have properly improved the manuscript, that I consider of interest for Readers working in the field.